# Guided Trajectory Generation with Diffusion Models for Offline Model-based Optimization

**Taeyoung Yun**[1]   **Sujin Yun**[1]   **Jaewoo Lee**[1]   **Jinkyoo Park**[1,2]
[1]Korea Advanced Institute of Science and Technology (KAIST)   [2]Omelet.ai
`{99yty, yunsj0625, jaewoo, jinkyoo.park}@kaist.ac.kr`

## Abstract

Optimizing complex and high-dimensional black-box functions is ubiquitous in science and engineering fields. Unfortunately, the online evaluation of these functions is restricted due to time and safety constraints in most cases. In offline model-based optimization (MBO), we aim to find a design that maximizes the target function using only a pre-existing offline dataset. While prior methods consider forward or inverse approaches to address the problem, these approaches are limited by conservatism and the difficulty of learning highly multi-modal mappings. Recently, there has been an emerging paradigm of learning to improve solutions with synthetic trajectories constructed from the offline dataset. In this paper, we introduce a novel conditional generative modeling approach to produce trajectories toward high-scoring regions. First, we construct synthetic trajectories toward high-scoring regions using the dataset while injecting locality bias for consistent improvement directions. Then, we train a conditional diffusion model to generate trajectories conditioned on their scores. Lastly, we sample multiple trajectories from the trained model with guidance to explore high-scoring regions beyond the dataset and select high-fidelity designs among generated trajectories with the proxy function. Extensive experiment results demonstrate that our method outperforms competitive baselines on Design-Bench and its practical variants. The code is publicly available in `https://github.com/dbsxodud-11/GTG`.

## 1   Introduction

Optimizing complex and high-dimensional black-box functions is ubiquitous in science and engineering fields, including biological sequence design [1], materials discovery [2], and mechanical design [3, 4]. Traditional methods like Bayesian optimization have been developed to solve the problem by iteratively querying a black-box function. However, the online evaluation of the black-box function is restricted in most real-world situations due to time and safety constraints.

Fortunately, we often have access to a previously collected offline dataset. This problem setting is referred to as offline model-based optimization (MBO), and our objective is to find a design that maximizes a target function using solely an offline dataset [5]. As no online evaluation is available, a key challenge of MBO is the out-of-distribution (OOD) issue arising from limited data coverage. Suppose we train a proxy that predicts function values given input designs and naively apply a gradient-based optimizer based on the proxy to identify the optimal design. It would fall into sub-optimal results due to inaccurate predictions of the proxy in unseen regions.

To mitigate this issue, forward approaches mostly consider training a robust surrogate model against adversarial optimization of inputs and applying gradient-based maximization. Trabucco et al. [6] train a proxy with the regularization term to prevent overestimation on OOD designs. Fu and Levine [7] leverage normalized maximum likelihood estimator to handle uncertainty on unseen regions.

There are also several works that focus on fine-tuning the proxy for robustness on unexplored regions [8, 9, 10]. However, the generalization of the proxy outside of the dataset still remains challenging.

On the other hand, inverse approaches learn a mapping from function values to the input domain. Then, they generate high-scoring designs by querying the learned mapping with a high score. Prior approaches utilize expressive generative models to learn a mapping, such as variational autoencoders [11, 12], generative adversarial nets [13], autoregressive models [14] or diffusion models [15]. While these methods show promising results, they still suffer from the difficulty of learning highly unsmooth distributions and utilizing valuable information about the landscape of the black-box function.

Recently, a new perspective has emerged on tackling the MBO by learning to improve solutions with synthetic trajectories constructed from the dataset [16, 17]. These methods aim to generate a sequence of designs toward high-scoring regions. It seems more promising than learning an inverse mapping that generates only a single design, as we can utilize information from sequences of designs that can help better understand the landscape of the target function. However, there is still room for improvement in this perspective. First, prior approaches construct trajectories with simple heuristics, which may lead to generating trajectories with inconsistent directions of improvement. Furthermore, the sequential nature of autoregressive models may lead to error accumulation during sampling [18].

To this end, we propose a novel conditional generative modeling approach to solve the MBO problem. Unlike prior inverse approaches, which generate a single design, we generate a sequence of designs toward high-scoring regions with guided sampling. Our method consists of four stages. First, we construct trajectories from the dataset while incorporating locality bias to distill the knowledge of the landscape of the target function into the generator. Then, we train a conditional diffusion model that generates the whole trajectory at once to bypass error accumulation and an auxiliary proxy. After training, we sample multiple trajectories conditioned on context data points and high score values. Finally, we select high-fidelity designs among generated trajectories by filtering with the proxy.

We empirically demonstrate that our method achieves superior performance on Design-Bench, a well-known benchmark for MBO with a variety of real-world tasks. Furthermore, we explore more practical settings, such as sparse or noisy datasets, verifying the generalizability of our method.

## 2 Preliminaries

### 2.1 Problem setup

In offline model-based optimization (MBO), we aim to find a design $\mathbf{x}$ that maximizes the target black-box function $f$. Unlike the typical black-box optimization setting, we can only access an offline dataset $\mathcal{D}$, and online evaluations are unavailable. The problem setup can be described as follows:

$$\text{find } \mathbf{x}^* = \arg \max_{\mathbf{x} \in \mathbb{R}^d} f(\mathbf{x}) \text{ s.t only an offline dataset } \mathcal{D} = \{(\mathbf{x}_i, y_i)\}_{i=1}^N \text{ is given} \tag{1}$$

where $\mathbf{x}$ is a decision variable and $y = f(\mathbf{x})$ is a target property we want to maximize.

### 2.2 Diffusion probabilistic models

Diffusion probabilistic models [19, 20] are a class of generative models that approximate the true distribution $q_0$ with a parametrized model of the form: $p_\theta(x_0) = \int p_\theta(x_{0:T}) dx_{1:T}$, where $x_0 \sim q_0$ and $x_1, \cdots, x_T$ are latents with the same dimensionality. The joint distribution $p_\theta(x_{0:T})$ is called the reverse process, defined as a Markov chain starting from standard Gaussian $p_T(x_T) = \mathcal{N}(0, I)$:

$$p_\theta(x_{0:T}) = p_T(x_T) \prod_{t=1}^T p_\theta(x_{t-1}|x_t), \quad p_\theta(x_{t-1}|x_t) = \mathcal{N}(\mu_\theta(x_t, t), \Sigma_t) \tag{2}$$

where $p_\theta(x_{t-1}|x_t)$ is parametrized Gaussian transition from timestep $t$ to $t-1$.

We define a forward process, which is also fixed as a Markov chain that adds Gaussian noise to the data with the variance schedule $\beta_1, \cdots, \beta_T$:

$$q(x_{1:T}|x_0) = \prod_{t=1}^T q(x_t|x_{t-1}), \quad q(x_t|x_{t-1}) = \mathcal{N}(\sqrt{1-\beta_t}x_{t-1}, \beta_t I) \tag{3}$$

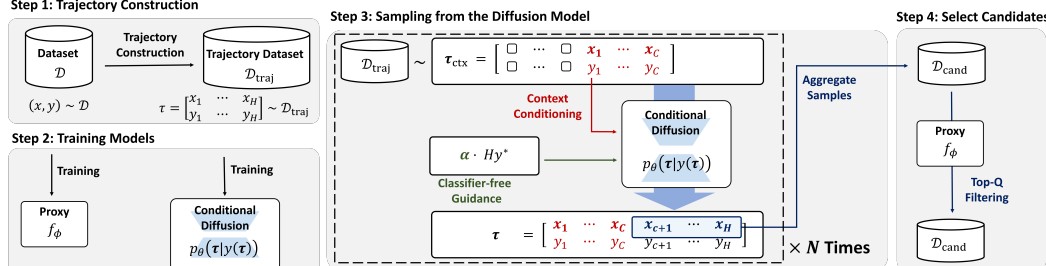

Figure 1: Overview of our method. **Step 1:** Construct trajectories from the dataset. **Step 2:** Train diffusion model and proxy. **Step 3:** Sample trajectories from the diffusion model with classifier-free guidance and context conditioning. **Step 4:** Select candidates for evaluation by filtering with proxy.

Training diffusion models can be performed by maximizing the variational lower bound on the log-likelihood $\mathbb{E}_{q_0}\left[\log p_\theta(x_0)\right]$, which is equivalent to minimizing the following loss:

$$\mathcal{L}(\theta) = \mathbb{E}_{x_0 \sim q_0, t \sim U(1,T), \epsilon \sim \mathcal{N}(0,I)}\left[\|\epsilon - \epsilon_\theta(x_t, t)\|^2\right] \tag{4}$$

where $\epsilon_\theta(x_t, t)$ is the parameterization suggested by [20], $\mu_\theta(x_t, t) = \frac{1}{\sqrt{\alpha_t}}\left(x_t - \frac{\beta_t}{\sqrt{1-\bar{\alpha}_t}}\epsilon_\theta(x_t, t)\right)$.

For modeling conditional distribution $q(x|y)$, we can use classifier-free guidance [21]. In classifier-free guidance, we train both a conditional $\epsilon_\theta(x_t, y, t)$ and unconditional model $\epsilon_\theta(x_t, t)$ with the following loss:

$$\mathcal{L}(\theta) = \mathbb{E}_{x_0, y \sim q(x,y), t \sim U(1,T), \epsilon \sim \mathcal{N}(0,I), \beta \sim \text{Bern}(p)}\left[\|\epsilon - \epsilon_\theta(x_t, (1-\beta)y + \beta\emptyset, t)\|^2\right] \tag{5}$$

For sampling, we start from Gaussian noise $x_T$ and refine $x_t$ into $x_{t-1}$ with the perturbed noise from the learned model $\epsilon_\theta$ at each diffusion timestep $t$:

$$\hat{\epsilon}(t) = \epsilon_\theta(x_t, \emptyset, t) + \omega(\epsilon_\theta(x_t, y, t) - \epsilon_\theta(x_t, \emptyset, t)) \tag{6}$$

where $\omega$ is a scalar value that controls the guidance scale.

## 3 Methodology

In this section, we introduce **GTG**, **G**uided **T**rajectory **G**eneration, a conditional generative modeling approach for solving MBO problem by learning to improve solutions using the offline dataset. We first construct trajectories towards high-scoring regions while incorporating locality bias for consistent improvement directions. Then, we train the conditional diffusion model to generate trajectories and a proxy model. Finally, we sample multiple trajectories using the diffusion model with guided sampling and filter high-fidelity designs with the proxy. Figure 1 shows the overview of the proposed method.

### 3.1 Constructing trajectories

We construct a set of trajectories $\mathcal{D}_{\text{traj}}$ from the offline dataset $\mathcal{D}$ to gather information on learning to improve designs. In this paper, each trajectory $\tau \in \mathcal{D}_{\text{traj}}$ is a set of $H$ input-output pairs and can be represented as a two-dimensional array:

$$\tau = \begin{bmatrix} \mathbf{x}_1 & \mathbf{x}_2 & \cdots & \mathbf{x}_H \\ y_1 & y_2 & \cdots & y_H \end{bmatrix}, \quad (\mathbf{x}_h, y_h) \in \mathcal{D} \ \forall h = 1, \cdots, H \tag{7}$$

While prior works construct trajectories via sorting heuristics or sampling from high-scoring regions, we focus on constructing trajectories that give us more valuable information for learning to improve designs towards higher scores. To achieve this, we develop a novel method to construct trajectories based on two desiderata.

First, the trajectory should be towards high-scoring regions while containing information on the landscape of the target black-box function. Second, the trajectories should be diverse and not converge to a single data point with the highest score of the dataset, as our objective is to discover high-scoring designs beyond the offline dataset by generalizing the knowledge of learning to improve solutions.

---

**Algorithm 1** Trajectory construction procedure of GTG

---

**Input:** Offline dataset $\mathcal{D}$, Trajectory length $H$, Number of trajectories $N$, initial percentile $p$, number of nearest neighbors $K$, and perturbation coefficient $\epsilon$.
**Output:** $\mathcal{D}_{\text{traj}}$
 1: Initialize trajectory dataset $\mathcal{D}_{\text{traj}} \longleftarrow \emptyset$
 2: **for** $n = 1, \cdots, N$ **do**
 3:     Sample $(\mathbf{x}_1, y_1)$ from $p$th percentile of $\mathcal{D}$ and initialize trajectory $\boldsymbol{\tau} \longleftarrow \{(\mathbf{x}_1, y_1)\}$
 4:     **for** $h = 1, \cdots, H - 1$ **do**
 5:         Find $K$ nearest neighbors of $\mathbf{x}_h$ whose score is higher than $\max\{y_1, \cdots, y_h\} - \epsilon$
 6:         Sample $(\mathbf{x}_{h+1}, y_{h+1})$ from the $K$ neighbors and update $\boldsymbol{\tau} \longleftarrow \boldsymbol{\tau} \cup \{(\mathbf{x}_{h+1}, y_{h+1})\}$
 7:     **end for**
 8:     Update $\mathcal{D}_{\text{traj}} \longleftarrow \mathcal{D}_{\text{traj}} \cup \{\boldsymbol{\tau}\}$
 9: **end for**

---

To this end, we introduce a novel strategy to construct trajectories from the dataset. We illustrate the procedure in Algorithm 1. For each trajectory, we first sample an initial data point $(\mathbf{x}_1, y_1)$ from a relatively low score distribution, $p$th percentile of $\mathcal{D}$. After initialization, we employ a local search strategy to select the next data point to generate a smooth trajectory toward high-scoring regions that contain the information on the landscape of the target function. Specifically, for each round $h$, we find $K$ nearest neighbors of $\mathbf{x}_h$ whose score is higher than $\max\{y_1, \cdots, y_h\} - \epsilon$, where $\epsilon$ is a small, non-negative real number. By allowing small perturbations using $\epsilon$, we can prevent generated trajectories from converging a single maximum of the offline dataset. Then, we sample $(\mathbf{x}_h, y_h)$ from the $K$ neighbors randomly to generate diverse trajectories. We repeat the procedure until constructing a trajectory of length $H$. By moving towards high-scoring regions while staying in a local region, we can effectively guide the generator to learn diverse and consistent paths for improving solutions.

Note that identifying $K$ nearest neighbors of a data point whose values are above a certain threshold does not require substantial computational time compared to training and evaluation. We explain in more detail our trajectory construction procedure in Appendix B.1

## 3.2 Training models

Given our trajectory dataset $\mathcal{D}_{\text{traj}}$, our objective is to learn the conditional distribution of trajectories towards high-scoring regions. We choose diffusion models, which have a powerful capability to learn the distribution of complex and high-dimensional data [22, 23], to generate trajectories. Our objective is then transformed from searching high-scoring designs to maximizing the conditional likelihood of trajectories, which can be achieved by minimizing the loss in Equation (5):

$$\theta^* = \arg\max_{\theta} \mathbb{E}_{\boldsymbol{\tau} \sim \mathcal{D}_{\text{traj}}} \left[ \log p_\theta(\boldsymbol{\tau} | y(\boldsymbol{\tau})) \right] \tag{8}$$

where $y(\boldsymbol{\tau}) = \sum_{h=1}^{H} y_h$ is the sum of scores in the trajectory $\boldsymbol{\tau}$. By training a diffusion model to generate a sequence of designs instead of a single design, we can efficiently distill the knowledge of the complex landscape of the target function into the diffusion model.

In addition, we also train a forward proxy $f_\phi$ using the dataset $\mathcal{D}$. We can use the proxy to filter high-scoring designs from the trajectories generated by the trained diffusion model.

## 3.3 Sampling trajectories from the diffusion model

After training, we sample trajectories with guided sampling. We use *classifier-free guidance* to generate trajectories. To be specific, we sample $\boldsymbol{\tau}$ from the diffusion model using Equation (6), where $y^*(\boldsymbol{\tau})$ is the target conditioning value. Following prior works [13, 16], we assume that we know the maximum score $y^*$ and set $y^*(\boldsymbol{\tau}) = \alpha \cdot (Hy^*)$, where $\alpha$ controls the exploration level of the generated trajectories. We discuss the role of $\alpha$ in more detail in the subsequent section.

To fully utilize the expressive power of diffusion models, we introduce an additional strategy, *context conditioning*, during the sampling. We generate trajectory with diffusion model while inpainting the $C$ context data points of the trajectory with $\boldsymbol{\tau}_{\text{ctx}}$, which is a subtrajectory sampled from $\mathcal{D}_{\text{traj}}$. By conditioning trajectories in different contexts, we can effectively explore diverse high-scoring regions.

**Algorithm 2** Sampling procedure of GTG

**Input:** Offline dataset $\mathcal{D}$, Trajectory dataset $\mathcal{D}_{\text{traj}}$, Conditional diffusion model $p_\theta$, Proxy model $f_\phi$, Context length $C$, Trajectory length $H$, Evaluation budget $Q$.
**Output:** $\mathcal{D}_{\text{cand}}$
 1: Initialize $\mathcal{D}_{\text{cand}} \longleftarrow \emptyset$
 2: **for** $n = 1, \cdots, N$ **do**
 3:     Initialize $\boldsymbol{\tau}^{(T)} \sim \mathcal{N}(0, I)$ and $\boldsymbol{\tau}_{\text{ctx}} \sim \mathcal{D}_{\text{traj}}$
 4:     **for** $t = T, \cdots, 1$ **do**
 5:         Compute $\hat{\epsilon}(t)$ using $p_\theta(\boldsymbol{\tau}|y^*(\boldsymbol{\tau}))$ by Equation (6)         $\triangleright$ *Classifier-free Guidance*
 6:         Compute $\boldsymbol{\tau}^{(t-1)}$ using $\boldsymbol{\tau}_{\text{ctx}}$, $\hat{\epsilon}(t)$ and $\boldsymbol{\tau}^{(t)}$ by Equation (9)         $\triangleright$ *Context-Conditioning*
 7:     **end for**
 8:     Update $\mathcal{D}_{\text{cand}} \longleftarrow \mathcal{D}_{\text{cand}} \cup \{\mathbf{x}_{C+1}, \cdots, \mathbf{x}_H\}$ from $\boldsymbol{\tau}(= \boldsymbol{\tau}^{(\mathbf{0})})$
 9: **end for**
10: Set $\mathcal{D}_{\text{cand}}$ as top-$Q$ scoring samples filtering by $f_\phi$         $\triangleright$ *Filtering*

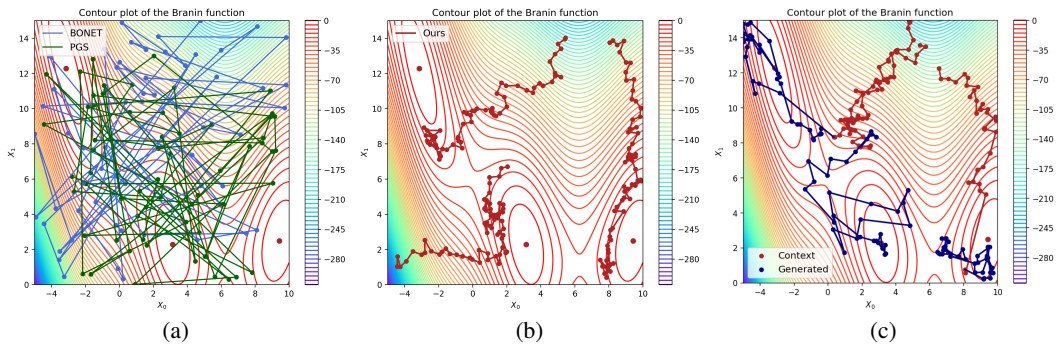

Figure 2: (a) Trajectories constructed by BONET (blue) and PGS (green). (b) Diverse trajectories constructed by GTG (red). (c) Trajectories generated by trained diffusion model with guided sampling. Red dots indicate context data points, and blue dots represent generated data points.

Formally, for each denoising timestep $t$, we refine $\boldsymbol{\tau}^{(t)}$ into $\boldsymbol{\tau}^{(t-1)}$ with the following procedure:

$$\boldsymbol{\tau}^{(t-1)} = \mathbf{m} \odot \boldsymbol{\tau}_{\text{ctx}} + (1 - \mathbf{m}) \odot \frac{1}{\sqrt{\alpha_t}} \left( \boldsymbol{\tau}^{(t)} - \frac{\beta_t}{\sqrt{1 - \bar{\alpha}_t}} \hat{\boldsymbol{\epsilon}}(t) \right) \tag{9}$$

where $\mathbf{m}$ is the mask for the first $C$ context data points and $\hat{\boldsymbol{\epsilon}}(t)$ is computed from the Equation (6).

### 3.4 Selecting candidates

After generating trajectories, we introduce *filtering* to select candidates for evaluation. In other words, we select top-$Q$ samples in terms of the predicted score from the proxy. By filtering with the proxy, we can exploit the knowledge from the dataset to search high-scoring designs [13, 14, 24].

## 4 Experimental evaluation

In this section, we present the results of our experiments on various tasks. First, we analyze our method in a toy 2D experiment. Then, we present the results on the Design-Bench and its practical variants to verify the effectiveness of the method. We also conduct extensive analyses on various aspects to deepen our understanding of the proposed method.

### 4.1 Toy 2D experiment

We first evaluate our method using a toy setting to analyze each component of our method thoroughly. We choose Branin, a synthetic 2D function with three distinct global maxima. Figure 2 shows the contour plot of the Branin function. The analytical form of the Branin function is as follows:

$$f(x_1, x_2) = -a \left( x_2 - bx_1^2 + cx_1 - r \right)^2 - s \left( 1 - t \right) \cos(x_1) - s \tag{10}$$

Table 1: Experiments on Design-Bench Tasks. We report max score ($100^{th}$ percentile) among $Q$=128 candidates. **Blue** denotes the best entry in the column, and **Violet** denotes the second best.

| Method | TFBind8 | TFBind10 | Superconductor | Ant | D'Kitty | Mean Rank |
|--------|---------|----------|----------------|-----|---------|-----------|
| $\mathcal{D}$ (best) | 0.439 | 0.467 | 0.399 | 0.565 | 0.884 | - |
| BO-qEI | 0.794 ± 0.103 | 0.631 ± 0.041 | 0.486 ± 0.025 | 0.812 ± 0.000 | 0.896 ± 0.000 | 11.4 / 15 |
| CMA-ES | 0.919 ± 0.055 | 0.649 ± 0.020 | 0.478 ± 0.010 | **2.222 ± 1.550** | 0.724 ± 0.001 | 8.6 / 15 |
| REINFORCE | 0.947 ± 0.029 | 0.628 ± 0.025 | 0.485 ± 0.011 | 0.247 ± 0.031 | 0.558 ± 0.193 | 11.6 / 15 |
| Grad Ascent | **0.983 ± 0.015** | 0.648 ± 0.044 | 0.509 ± 0.018 | 0.295 ± 0.021 | 0.877 ± 0.023 | 7.8 / 15 |
| COMs | 0.968 ± 0.025 | 0.619 ± 0.038 | 0.444 ± 0.035 | 0.927 ± 0.043 | 0.957 ± 0.016 | 8.2 / 15 |
| NEMO | 0.941 ± 0.000 | **0.705 ± 0.000** | 0.502 ± 0.002 | 0.952 ± 0.002 | 0.950 ± 0.001 | **4.8 / 15** |
| RoMA | 0.924 ± 0.040 | 0.666 ± 0.035 | **0.510 ± 0.015** | 0.917 ± 0.030 | 0.927 ± 0.013 | 7 / 15 |
| BDI | 0.973 ± 0.000 | 0.630 ± 0.025 | 0.508 ± 0.011 | 0.932 ± 0.000 | 0.939 ± 0.000 | 6.6 / 15 |
| ICT | 0.944 ± 0.015 | 0.598 ± 0.020 | 0.507 ± 0.014 | 0.946 ± 0.021 | **0.960 ± 0.014** | 7 / 15 |
| CbAS | 0.895 ± 0.043 | 0.638 ± 0.040 | 0.468 ± 0.058 | 0.825 ± 0.030 | 0.898 ± 0.011 | 11.2 / 15 |
| MINs | 0.884 ± 0.039 | 0.660 ± 0.048 | 0.500 ± 0.036 | 0.908 ± 0.031 | 0.942 ± 0.005 | 8.8 / 15 |
| DDOM | 0.966 ± 0.015 | 0.666 ± 0.024 | 0.476 ± 0.029 | 0.926 ± 0.027 | 0.948 ± 0.011 | 7 / 15 |
| BONET | 0.831 ± 0.109 | 0.606 ± 0.044 | 0.405 ± 0.017 | 0.957 ± 0.004 | 0.950 ± 0.014 | 10 / 15 |
| PGS | 0.968 ± 0.019 | 0.693 ± 0.031 | 0.475 ± 0.048 | 0.748 ± 0.049 | 0.948 ± 0.014 | 7.6 / 15 |
| **GTG (Ours)** | **0.976 ± 0.020** | **0.698 ± 0.127** | **0.519 ± 0.045** | **0.963 ± 0.009** | **0.971 ± 0.009** | **1.6 / 15** |

where $a = 1$, $b = \frac{5.1}{4\pi^2}$, $c = \frac{5}{\pi}$, $s = 10$, $t = \frac{1}{8\pi}$ and the range of $(x_1, x_2)$ is $[-5, 10] \times [0, 15]$.

For the MBO setting, we uniformly sample 5000 data points and remove the top 10% percentile to make the task more challenging. We construct trajectories with a length of 64 using our trajectory construction strategy and other strategies suggested by prior methods, BONET [16] and PGS [17].

Figure 2a shows the trajectories generated from prior methods. As shown in the figure, we find that constructed trajectories show uncorrelated movements, which makes the model hard to capture knowledge on the landscape of the target black-box function. Unlike prior methods, our method constructs trajectories that improve the solution with the local movements, as illustrated in Figure 2b. Such trajectories help the diffusion model to learn how to improve solutions efficiently. We also find that trajectories do not converge into a single data point and toward diverse high-scoring regions via random sampling from $K$ neighbors and perturbations from $\epsilon$.

Figure 2c shows the trajectories generated by the trained diffusion model with context conditioning and classifier-free guidance. As shown in the figure, GTG can generalize the knowledge on improving solutions to find diverse high-scoring solutions. GTG achieves a maximum score of $-0.490 \pm 0.070$, which is near-optimal compared to the global optimum ($-0.398$) and far beyond the maximum value of the dataset ($-6.031$). Please refer to Appendix A.1 for more details of the toy experiment.

## 4.2 Design-Bench tasks

In this section, we present the experiment results of our method on Design-Bench tasks [5]. We conduct experiments on two discrete tasks and three continuous tasks. For each task, we have an offline dataset from an unknown oracle function. We present the detailed task description below.

**TFBind8 and TFBind10 [1].** We aim to find a DNA sequence of lengths 8 and 10 with maximum binding affinity with a particular transcription factor.

**Superconductor [2].** We aim to design a chemical formula, represented by an 86-dimensional vector, for a superconducting material with a high critical temperature.

**Ant and D'Kitty Morphology [4, 25].** We aim to optimize the morphological structure of two simulated robots. The morphology parameters include size, orientation, and the location of the limbs. Ant has 60 continuous parameters, and D'Kitty has 56 continuous parameters.

## 4.3 Baselines

For baselines, we prepare four main categories to solve MBO problems. First, we compare our method with traditional methods widely used in online black-box optimization settings, such as BO-qEI [26], CMA-ES [27], REINFORCE [28], and Gradient Ascent.

Table 2: Experiments on Sparse Datasets.

| Method | TFBind8 | | | Dkitty | | |
|---|---|---|---|---|---|---|
| | 1% | 20% | 50% | 1% | 20% | 50% |
| BDI | 0.898 ± 0.000 | 0.952 ± 0.000 | **0.988 ± 0.000** | 0.865 ± 0.000 | 0.927 ± 0.000 | 0.938 ± 0.000 |
| ICT | 0.899 ± 0.045 | 0.925 ± 0.035 | 0.962 ± 0.019 | 0.946 ± 0.010 | 0.949 ± 0.010 | 0.954 ± 0.008 |
| DDOM | 0.851 ± 0.082 | 0.906 ± 0.050 | 0.896 ± 0.048 | 0.938 ± 0.007 | 0.945 ± 0.011 | 0.944 ± 0.008 |
| BONET | 0.791 ± 0.079 | 0.824 ± 0.061 | 0.884 ± 0.072 | 0.875 ± 0.004 | 0.939 ± 0.007 | 0.940 ± 0.009 |
| **GTG (Ours)** | **0.948 ± 0.009** | **0.964 ± 0.025** | 0.973 ± 0.016 | **0.949 ± 0.013** | **0.957 ± 0.009** | **0.968 ± 0.002** |

Table 3: Experiments on Noisy Datasets.

| Method | TFBind8 | | | Dkitty | | |
|---|---|---|---|---|---|---|
| | 1% | 20% | 50% | 1% | 20% | 50% |
| BDI | **0.980 ± 0.005** | 0.886 ± 0.051 | 0.873 ± 0.048 | 0.929 ± 0.008 | 0.908 ± 0.010 | 0.918 ± 0.016 |
| ICT | 0.941 ± 0.013 | 0.950 ± 0.023 | 0.921 ± 0.054 | 0.940 ± 0.029 | 0.914 ± 0.024 | 0.896 ± 0.000 |
| DDOM | 0.896 ± 0.048 | 0.887 ± 0.065 | 0.887 ± 0.065 | 0.944 ± 0.009 | 0.945 ± 0.011 | 0.926 ± 0.020 |
| BONET | 0.904 ± 0.044 | 0.822 ± 0.113 | 0.773 ± 0.143 | 0.942 ± 0.008 | 0.927 ± 0.024 | 0.924 ± 0.010 |
| **GTG (Ours)** | 0.976 ± 0.015 | **0.967 ± 0.026** | **0.948 ± 0.029** | **0.955 ± 0.008** | **0.947 ± 0.015** | **0.937 ± 0.013** |

The second category comprises recently proposed forward approaches, including COMs [6], NEMO [7], RoMA [8], BDI [24], and ICT [9]. The third category encompasses inverse approaches, and we select CbAS [11], MINs [13], and DDOM [15] as our baselines. Finally, we also compare with baselines which construct synthetic trajectories and generalize the knowledge of learning to improve solutions, BONET [16] and PGS [17].

## 4.4 Evaluation metrics

For evaluation, we follow the protocol of prior works. We identify $Q = 128$ designs selected by the algorithm and report a normalized score of $100^{th}$ percentile design. For all algorithms, we run experiments over 8 different seeds and report mean and standard errors.

To evaluate our method, we construct trajectories of length $H = 64$ and train a conditional diffusion model for each task. After training, we sample $N = 128$ trajectories conditioning on $C = 32$ context data points and setting $\alpha = 0.8$ across all tasks. Finally, we filter top-128 candidates among generated designs with the predicted score from the proxy for evaluation.

## 4.5 Main results

As shown in the Table 1, GTG achieves an average rank of 1.6, the best among all competitive baselines. It performs best on two tasks and is runner-up on three tasks, demonstrating superior performance across different tasks. We observe that GTG generally surpasses forward approaches, which struggle to fall into OOD designs, especially in high-dimensional settings. We also observe that our method outperforms inverse approaches, including DDOM, which also utilizes a diffusion model. It demonstrates that generating trajectories towards high-scoring regions can be more effective than generating a single design, as we can distill the knowledge of the landscape of the target function into the generator. Our method achieves higher performance compared to BONET, which also generates trajectories. It indicates that our novel trajectory construction strategy effectively guides the diffusion model to explore diverse paths toward high-scoring regions.

## 4.6 Practical variants of Design-Bench tasks

In this section, we present experiment results in a more practical setting of Design-Bench tasks. While Design-Bench assumes a large, unbiased offline dataset containing thousands of data points for the training model, such a setting is impractical in most cases. Therefore, we prepare two additional practical settings, sparse and noisy datasets, to verify the robustness of our method in such extreme cases. In a sparse setting, we only provide $x\%$ of the original dataset for training. For the noisy setting, we add $x\%$ of standard Gaussian noise to the normalized score values. We choose recent papers published after 2022, BDI, ICT, DDOM, and BONET for primary baselines. Please refer to Appendix A.2 for detailed experiment settings and Appendix D.5 for results with more baselines.

Table 4: Ablation study on trajectory construction strategy.

| Method | TFBind8 | TFBind10 | Superconductor | Ant | D'Kitty |
|---|---|---|---|---|---|
| SORT-SAMPLE | $0.954 \pm 0.026$ | $0.697 \pm 0.126$ | $0.487 \pm 0.016$ | $0.946 \pm 0.011$ | $0.966 \pm 0.005$ |
| Top-$p$ Percentile | $0.948 \pm 0.030$ | $0.669 \pm 0.033$ | $0.439 \pm 0.039$ | $0.946 \pm 0.018$ | $0.964 \pm 0.003$ |
| Ours | $\textbf{0.976} \pm \textbf{0.020}$ | $\textbf{0.698} \pm \textbf{0.127}$ | $\textbf{0.519} \pm \textbf{0.045}$ | $\textbf{0.963} \pm \textbf{0.009}$ | $\textbf{0.971} \pm \textbf{0.009}$ |

Table 5: Ablation study on sampling procedure of GTG.

| Method | TFBind8 | TFBind10 | Superconductor | Ant | D'Kitty |
|---|---|---|---|---|---|
| $\emptyset$ | $0.923 \pm 0.054$ | $0.636 \pm 0.047$ | $0.499 \pm 0.040$ | $0.867 \pm 0.051$ | $0.926 \pm 0.048$ |
| $\{\text{CF}\}$ | $0.914 \pm 0.053$ | $0.687 \pm 0.065$ | $0.502 \pm 0.040$ | $0.918 \pm 0.064$ | $0.943 \pm 0.011$ |
| $\{\text{CF}, \text{CC}\}$ | $0.920 \pm 0.036$ | $0.687 \pm 0.065$ | $0.502 \pm 0.024$ | $0.927 \pm 0.022$ | $0.945 \pm 0.014$ |
| $\{\text{CF}, \text{F}\}$ | $0.963 \pm 0.019$ | $0.628 \pm 0.036$ | $0.483 \pm 0.034$ | $0.952 \pm 0.026$ | $0.965 \pm 0.007$ |
| $\{\text{CF}, \text{CC}, \text{F}\}$ | $\textbf{0.976} \pm \textbf{0.020}$ | $\textbf{0.698} \pm \textbf{0.127}$ | $\textbf{0.519} \pm \textbf{0.045}$ | $\textbf{0.963} \pm \textbf{0.009}$ | $\textbf{0.971} \pm \textbf{0.009}$ |

Table 2 shows the results of our method and recent baselines in sparse datasets. The table shows that our method mostly outperforms other baselines even in sparse datasets, demonstrating the superiority of exploiting knowledge of the target function by constructing diverse trajectories from the dataset. Table 3 reports the experiment results on the noisy settings. We find that even with 50% of noise, our method can find relatively high-scoring designs, demonstrating its robustness in practical settings.

## 5   Additional analysis

In this section, we carefully analyze the effectiveness of each component in our method.

**Ablation on trajectory construction.** We propose a novel trajectory construction strategy by incorporating locality bias. To verify the effectiveness of the strategy, we compare our strategy with prior approaches, SORT-SAMPLE and Top-$p$ Percentile, suggested by BONET and PGS, respectively. Table 4 shows that our strategy outperforms prior strategies across various tasks. We conduct additional analysis on trajectory construction strategies in Appendix D.1.

**Ablation on sampling procedure.** We analyze the effectiveness of strategies we introduced during the sampling procedure, namely context conditioning (CC), classified-free guidance (CF), and filtering (F). Across various tasks, it is evident that all components are crucial for improving performance as demonstrated in Table 5. We conduct further analysis on sampling strategies in Appendix D.2.

**Hyperparameter sensitivity.** We also conduct experiments on the effect of various hyperparameters we introduced in this paper. We first train a conditional diffusion model with various lengths ($H$). As shown in Figure 3a, increasing $H$ leads to achieving higher performance. We also conduct experiments by varying the number of contexts ($C$) and the exploration level ($\alpha$). Figure 3b shows that $C = 32$ achieves superior performance while conditioning with too many contexts degrades performance. Finally, Figure 3b shows a strong correlation between $\alpha$ and the score, demonstrating the effectiveness of guided sampling. We conduct further analysis on hyperparameters in Appendix D.2.

**Varying evaluation budget.** We provide experiment results with a small number of evaluation budgets ($Q$). As shown in Figure 4, we generally outperform most baselines even with a relatively low evaluation budget.

**Assumption on optimal score.** We assume that the optimal value $y^*$ is known, following prior works [13, 16]. We conduct experiments by relaxing the aforementioned assumption in Appendix D.2 and find that GTG can achieve comparable performance even without knowing $y^*$.

**Effect of unsupervised pretraining.** It might be beneficial to pretrain the diffusion model when we have a large-scale unlabeled dataset and a few designs of labeled points [29]. To this end, we discuss the effectiveness of pretraining diffusion models with unlabeled datasets in Appendix D.3.

**Time complexity of sampling procedure.** We also conduct analysis on the time complexity of the sampling procedure of our method in Appendix D.4. Experiment results demonstrate that we can decrease the number of denoising timesteps even one-tenth with minimal loss in performance.

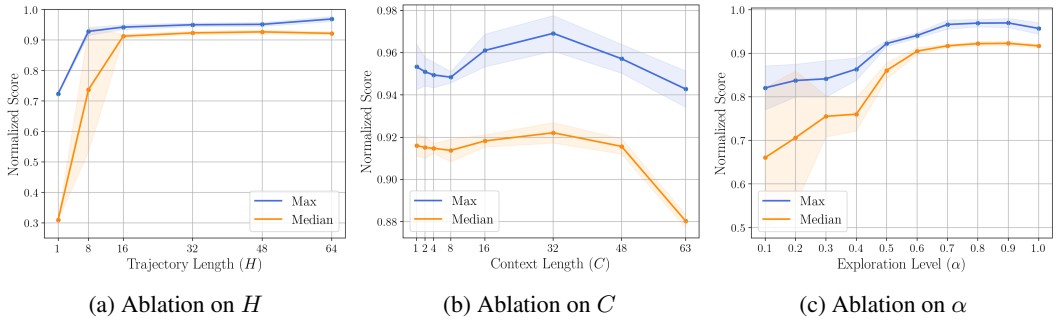

(a) Ablation on $H$      (b) Ablation on $C$      (c) Ablation on $\alpha$

Figure 3: Ablation on hyperparameters of GTG. Experiments are conducted on D'Kitty task.

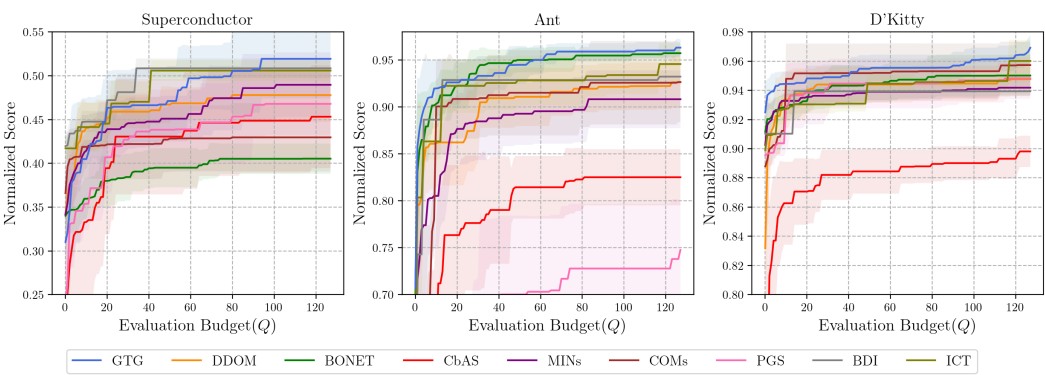

Figure 4: Ablation on varying evaluation budget $Q$.

# 6 Related works

## 6.1 Offline model-based optimization

In offline MBO, generalization outside the offline dataset is crucial for success. While there have been attempts to train a robust surrogate model to achieve accurate predictions on unseen regions [8, 9, 10], effectively exploring high-scoring regions remains challenging.

Recently, a new perspective on solving the MBO problem has emerged by learning to improve solutions from synthetic trajectories and generalizing the knowledge to find designs beyond the dataset [16, 17]. BONET [16] trains an autoregressive model to generate optimal trajectories conditioned on a low regret budget. PGS [17] trains RL policy with trajectories consisting of high-scoring designs to roll out optimal trajectories. MATCH-OPT [30] also constructs monotonic trajectories and matches the gradient field with the proxy. GTG falls under this category but adopts a unique approach to constructing trajectories with local search and utilizing diffusion models to enhance performance.

## 6.2 Generative models for decision making

Generative models have emerged as a powerful tool for decision-making problems, including bandit problems [31], reinforcement learning [18, 32, 33, 34, 35], and optimization [15, 36]. In offline MBO, there are inverse approaches to learning a mapping from function values to input domains with generative models and sample designs from high-scoring regions [11, 12, 14, 15]. DDOM [15] utilizes a conditional diffusion model and generates high-scoring samples with reweighted training and classifier-free guidance. DiffOPT [36] considers a constrained optimization setting and introduces a two-stage framework that begins with a guided diffusion process for warm-up, followed by a Langevin dynamics stage for further correction.

As concurrent works, DEMO [37] trains a diffusion model to match a pseudo-target distribution constructed by gradient ascent and uses the model to edit designs in the offline dataset. Diff-BBO [38] measures the uncertainty of generated designs to select the optimal target value for conditioning the diffusion model. Our method distinguishes itself from prior works by utilizing diffusion models to generate trajectories toward high-scoring regions by learning to improve solutions from the dataset.

# 7 Discussion and conclusion

In this paper, we introduce GTG, a novel conditional generative modeling approach for learning to improve solutions from synthetic trajectories constructed with the dataset. First, we construct diverse trajectories toward high-scoring regions while incorporating locality bias. Then, we train the conditional diffusion model and proxy function. After training, we generate trajectories with classifier-free guidance and context-conditioning to generalize the knowledge on how to improve solutions. Lastly, our filtering strategy for selecting candidates further improves the performance. Our extensive experiments demonstrate the generalizability of GTG.

**Limitation and future work.** While our method shows powerful generalizability on Design-Bench tasks, some evaluation methods may not fully capture real-world complexities. For example, in the superconductor task, we find that the offline dataset has multiple copies of the same inputs but with different outputs. As a result, the random forest oracle which is fit on this offline data is not reliable. Moreover, we resort to filtering designs with the proxy function, which may result in inaccurate predictions on OOD regions. Although our filtering strategy works well in sparse and noisy settings, one may consider constructing a robust proxy model to handle the uncertainty of its predictions.

# Acknowledgements

We thank the anonymous reviewers for their insightful comments and suggestions which significantly improve our manuscript. This work was supported by the National Research Foundation of Korea(NRF) grant funded by the Korea government(MSIT) (No. RS-2024-00410082).

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

# Appendix

## A    Task Details

We present additional information on Branin and Design-Bench tasks.

### A.1    Toy Branin Task

Branin is a well-known synthetic function for benchmarking black-box optimization methods. It has three distinct global maxima, $(-\pi, 12.275)$, $(\pi, 2.275)$, and $(9.42478, 2.475)$ with a maximum value of $-0.398$. We create a synthetic offline dataset by uniform sample $N = 5000$ data points and remove the top 10% percentile. Figure 5 shows the visualization of the dataset used for evaluation.

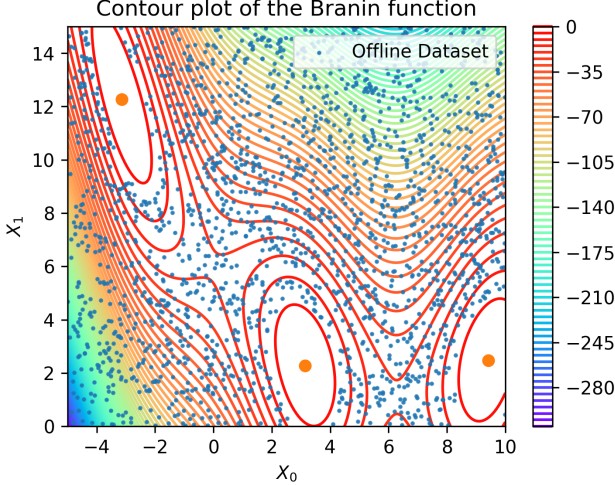

Figure 5: Visualization of the offline dataset used for Branin task.

We compare GTG with competitive baselines, BONET, and PGS for the Branin task. For all methods, we generate 400 trajectories with horizon 64 using construction strategies suggested by each method. For GTG, we train the diffusion model with a length $H = 64$ and apply context-conditioning with $C = 32$ and classifier-free guidance with $\alpha = 0.8$ for guided sampling. We generate four trajectories for evaluation. Table 6 shows the best function values achieved by each method on the Branin task. As shown in the table, GTG successfully generalizes the knowledge to improve solutions and achieve better performance compared to baselines.

Table 6: Experiment results on Branin task. We report 100th percentile among $Q = 128$ samples from each method. Experiments are conducted with three different random seeds.

| Optima | $\mathcal{D}$ (best) | BONET | PGS | GTG |
|---|---|---|---|---|
| -0.398 | -6.031 | -0.769 ± 0.227 | -1.295 ± 0.459 | **-0.490 ± 0.070** |

## A.2 Design-Bench Tasks

Design-Bench [5] is the most widely used benchmark for evaluating MBO algorithms. Table 7 shows the details of each task. For discrete tasks, we convert discrete input into a continuous vector by approximating logits with soft interpolation between one-hot encoding and uniform distribution using a mixing factor of $0.6$. We present detailed statistics of each task in Table 7.

Table 7: Detail Setting of Design-Bench Tasks.

| Task | Dataset Size | Dimensions | Type | Oracle | Max |
| --- | --- | --- | --- | --- | --- |
| TFBind8 | 32898 | 8 | Discrete | Exact | 1.0 |
| TFBind10 | 50000 | 10 | Discrete | Exact | 2.128 |
| Superconductor | 17014 | 86 | Continuous | Random Forest | 185.0 |
| Ant | 10004 | 60 | Continuous | Exact | 590.0 |
| Dkitty | 10004 | 56 | Continuous | Exact | 340.0 |

### A.2.1 Excluded Design-Bench Tasks

Following from prior works [15, 16, 29], we exclude Hopper [25] and ChEMBL [39] tasks for evaluation. As noted in previous works, the oracle for the Hopper task is heavily skewed towards low-function values and gives inconsistent results. For the ChEMBL task, all methods already produce nearly the same results, which makes it not a meaningful task for evaluation.

### A.2.2 Practical Variants of Design-Bench Tasks

We prepare two practical variants of Design-Bench tasks to verify the robustness of GTG in terms of data sparsity and label noise. We present the distribution of function values in the original offline dataset and its practical variants on the TFBind8 task and D'Kitty tasks. As shown in the figure, the score distributions of sparse and noisy datasets significantly differ from the original ones, making the task more challenging.

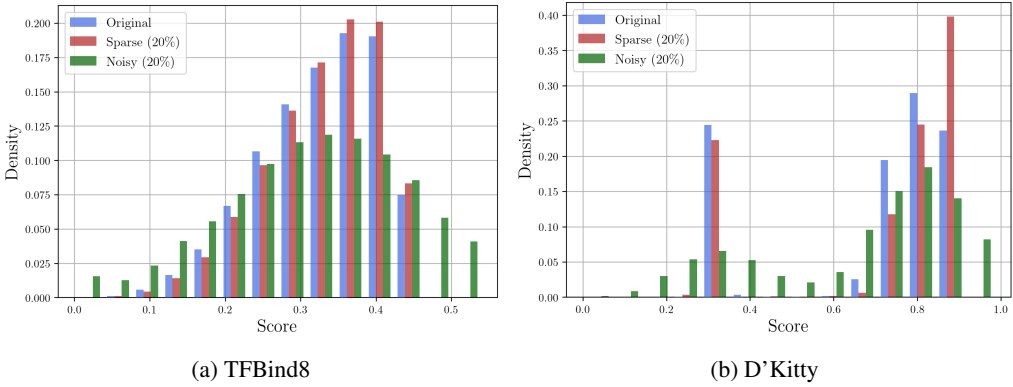

(a) TFBind8  (b) D'Kitty

Figure 6: Distribution of function values in the original offline dataset and its pratical variants.

# B  Methodology Details

In this section, we present the method details, including model implementations and architectures, training schemes, hyperparameter configurations, and computing resources.

## B.1  Trajectory Construction

In terms of constructing trajectories, we introduce two variables, $K$ and $\epsilon$, which control the level of locality and optimality of the trajectories. For too large value of $K$, we construct trajectories with inconsistent directions of improvement, while the extremely small value of $K$ leads to trajectories wandering the initial data point. If we lower the $\epsilon$ close to zero, we only allow monotonic improvement, while large $\epsilon$ values lead to suboptimal trajectories. We present the hyperparameters for our experiments in the Table 8. We also conduct additional analysis on trajectory construction in Appendix D.1.

Table 8: Hyperparameters for Trajectory Construction.

| Task | $p$ | $H$ | $N$ | $K$ | $\epsilon$ |
|------|-----|-----|------|-----|-----|
| TFBind8 | 20 | 64 | 1000 | 50 | 0.05 |
| TFBind10 | 20 | 64 | 1000 | 50 | 0.05 |
| Superconductor | 20 | 64 | 4000 | 20 | 0.05 |
| Ant | 20 | 64 | 4000 | 20 | 0.05 |
| Dkitty | 20 | 64 | 4000 | 20 | 0.01 |

To identify $K$ nearest neighbors of a certain data point, we pre-compute the distance matrix between pairwise designs. For discrete tasks, we use hamming-ball distance as a distance metric, and for continuous tasks, we use Euclidean distance to measure the similarity between designs. Table 9 shows the computational time for pre-computing distance matrix and constructing trajectory dataset from the offline dataset. As shown in the table, constructing trajectories does not require a significantly large amount of time, even in high-dimensional settings.

Table 9: Time complexity of trajectory construction on Design-Bench Tasks. We use Intel® Xeon® Gold 5317 CPU @ 3.00GHz and report mean and standard deviation across five different runs.

| Method | TFBind8 | TFBind10 | Superconductor | Ant | D'Kitty |
|--------|---------|----------|----------------|-----|---------|
| Distance Matrix (sec) | 5.38 ± 0.08 | 14.14 ± 2.17 | 7.34 ± 0.10 | 1.67 ± 0.01 | 1.49 ± 0.07 |
| Trajectory Construction (sec) | 22.36 ± 0.44 | 28.63 ± 0.26 | 73.19 ± 0.56 | 53.74 ± 3.05 | 56.24 ± 4.33 |

### B.2 Training Models

#### B.2.1 Training Diffusion Model

We use temporal U-Net architecture from Diffuser [18] as a backbone of the diffusion model. For discrete tasks, we train the model using Adam optimizer [40] for $1 \times 10^4$ training steps with the learning rate of $1 \times 10^{-3}$. While one could use discrete diffusion models [41, 42] for discrete tasks, we use continuous diffusion models with continuous relaxation of discrete inputs for simplicity. For continuous tasks, we train the model for $5 \times 10^4$ steps with a learning rate of $1 \times 10^{-4}$. The hyperparameters we used for modeling and training are listed in Table 10.

Table 10: Hyperparameters for Training Diffusion Models

|  | Parameters | Values |
|---|---|---|
| Architecture | Number of Layers | 6 |
|  | Num Channels | 32 (Discrete), 128 (Continuous) |
|  | Channel Multipliers | (1, 4, 8) |
| Training | Batch size | 128 |
|  | Optimizer | Adam |
|  | Learning Rate | $1 \times 10^{-3}$ (Discrete), $1 \times 10^{-4}$ (Continuous) |
|  | Training Steps | $1 \times 10^4$ (Discrete), $5 \times 10^4$ (Continuous) |
| Conditioning | Conditional dropout ($p$) | 0.25 |

#### B.2.2 Training Proxy Model

We use MLP with 2 hidden layers with 1024 hidden units and ReLU activations to implement the proxy function. As our objective is filtering high-fidelity designs with the proxy, we introduce a rank-based reweighting suggested by [43] during training to make the proxy model focus on high-scoring regions. For discrete tasks, we train a proxy model using Adam optimizer for $1 \times 10^3$ training steps with a learning rate of $1 \times 10^{-3}$. For continuous tasks, we train the model for $5 \times 10^3$ training steps with a learning rate of $1 \times 10^{-3}$. The hyperparameters we used for modeling and training are listed in Table 11.

Table 11: Hyperparameters for Training Proxy

|  | Parameters | Values |
|---|---|---|
| Architecture | Number of Layers | 2 |
|  | Num Units | 1024 |
| Training | Batch size | 128 |
|  | Optimizer | Adam |
|  | Learning Rate | $1 \times 10^{-3}$ |
|  | Training Steps | $1 \times 10^3$ (Discrete), $5 \times 10^3$ (Continuous) |

All training is done with a single NVIDIA RTX 3090 GPU and takes approximately 30 minutes for discrete tasks and 2 hours for continuous tasks.

### B.3 Sampling Procedure

We sample trajectories with $T = 200$ denoising steps across all tasks. For classifier-free guidance, we set the guidance scale $\omega$ as 1.2. In practice, we sample a batch of trajectories to generate multiple trajectories in parallel. We analyze the time complexity of sampling trajectories from the diffusion model in Appendix D.4

## C   Baseline Details

In this section, we provide more details on the baselines used for our experiments.

**Baselines from Design-Bench [5]**. We take the implementations of most baselines from open-source code[1]. It contains baselines of BO-qEI [26], CMA-ES [27], REINFORCE [28], Gradient Ascent, CbAS [11], MINs [13], and COMs [44]. We reproduce the results with 8 independent random seeds.

**NEMO [7]**. NEMO leverages a normalized maximum likelihood estimator to handle uncertainty in unseen regions and prevent adversarial optimization while performing gradient ascent. As there is no open-source code, we refer to the results of NEMO from [9].

**BDI [24]**. BDI learns forward mapping from low-scoring regions to high-scoring regions, and its backward mapping distills the knowledge of the offline dataset to search for optimal designs. We follow the hyperparameter setting of the paper and reproduce the results with the open-source code[2].

**ICT [9]**. ICT maintains three symmetric proxies and enhances the performance of the ensemble by co-teaching and importance-aware sample reweighting. We follow the hyperparameter setting of the paper and reproduce the results with the open-source code[3].

**DDOM [15]**. DDOM leverages diffusion models to model distribution over high-scoring regions and sample designs with classifier-free guidance. We follow the hyperparameter setting of the paper except for the evaluation budget $Q$ for a fair comparison. We find that there is a performance drop in several tasks when we use $Q = 128$ instead of $256$. We reproduce the results with the open-source code[4].

**BONET [16]**. BONET trains an autoregressive model with trajectories constructed from the offline dataset and generalizes the knowledge to explore high-scoring regions. We follow the hyperparameter setting of the paper except for the evaluation budget $Q$ for a fair comparison. We find that there is a performance drop in several tasks when we use $Q = 128$ instead of $256$. We reproduce the results with the open-source code[5].

**PGS [17]**. PGS trains a policy to guide gradient-based optimization by reformulating the MBO problem as an offline RL problem. We follow the hyperparameter setting of the paper and reproduce the results with the open-source code[6].

---

[1]https://github.com/brandontrabucco/design-baselines

[2]https://github.com/GGchen1997/BDI

[3]https://github.com/StevenYuan666/Importance-aware-Co-teaching

[4]https://github.com/siddarthk97/ddom

[5]https://github.com/siddarthk97/bonet

[6]https://github.com/yassineCh/PGS

# D Extended Additional Analysis

In this section, we present additional analysis on GTG which is not included in the main section due to the page limit.

## D.1 Additional Analysis on Trajectory Construction

### D.1.1 Analysis on Score Distribution of Trajectories

We conduct additional analysis on our trajectory construction method. We try to generate diverse trajectories toward high-scoring regions by randomly selecting subsequent designs from $K$ neighbors and allowing local perturbations. To this end, we visualize the shift in the distribution of function values via various trajectory construction strategies in the Superconductor task. As shown in Figure 7, the SORT-SAMPLE strategy suggested by BONET constructs trajectories solely on high-scoring designs, which can be easily trapped into local optima. Unlike SORT-SAMPLE, our method shifts distribution towards high-scoring regions while using the information of low-scoring regions to distill the knowledge of the landscape of the target function to the generator.

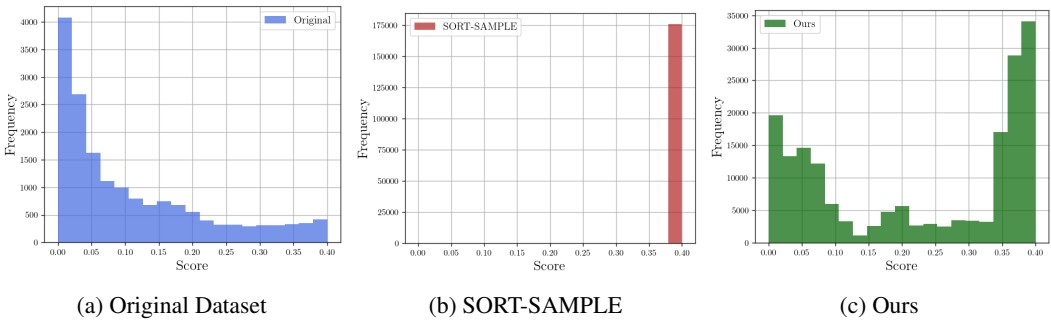

(a) Original Dataset      (b) SORT-SAMPLE      (c) Ours

Figure 7: Distribution of function values in the offline dataset and trajectory datasets constructed by different strategies.

### D.1.2 Analysis on Hyperparameters in Trajectory Construction

We also conduct additional analysis on hyperparameters in trajectory construction, $K$ and $\epsilon$. Figure 8 shows the performance of GTG in TFBind8 task by varying $K$ and $\epsilon$. While using too large $K$ or too small $\epsilon$ may lead to a relatively low performance, we do not see much variation with different values.

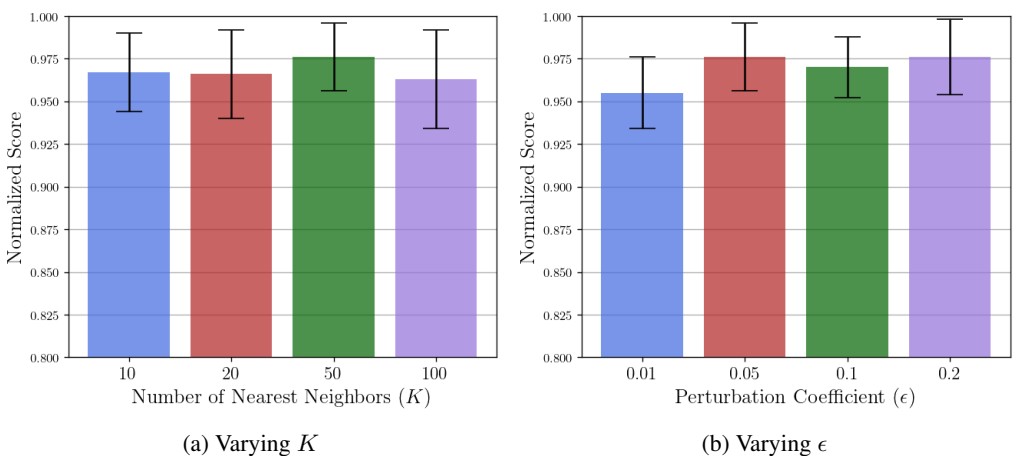

(a) Varying $K$      (b) Varying $\epsilon$

Figure 8: Performance of GTG in TFBind8 task by varying $K$ and $\epsilon$. Experiments are conducted with 8 random seeds and mean and standard deviation are reported.

### D.2 Additional Analysis on Sampling Procedure

### D.2.1 Various Strategies for Guided Sampling

In this section, we explore various strategies for guiding diffusion models to generate high-scoring designs. As we also generate score values, it could be possible to guide diffusion models to generate high-scoring designs by inpainting score values with the desired values. To this end, we conduct additional experiments on Design-Bench tasks by generating trajectories with inpainting instead of classifier-free guidance. Specifically, we inpaint the $y$ values of the generated trajectories as $y^*$, the normalized score of the optimal design.

Table 12 shows the performance of different guiding strategies. It confirms that conditioning by classifier-free guidance performs better than the inpainting strategy, justifying our decision choice.

Table 12: Exploring various guiding strategies.

| Method | TFBind8 | TFBind10 | Superconductor | Ant | D'Kitty |
|---|---|---|---|---|---|
| GTG (Inpainting) | 0.963 ± 0.026 | 0.652 ± 0.062 | 0.503 ± 0.035 | 0.938 ± 0.014 | 0.966 ± 0.007 |
| GTG (CF) | **0.976 ± 0.020** | **0.698 ± 0.127** | **0.519 ± 0.045** | **0.963 ± 0.009** | **0.971 ± 0.009** |

### D.2.2 Diversity Analysis

In this section, we explore the trade-off between performance and diversity via filtering strategy. While the filtering strategy boosts the performance of our method by eliminating potentially suboptimal designs, it may reduce the diversity of candidates, which may be crucial in tasks such as drug discovery due to proxy misspecification [45].

To this end, we measure the diversity of the candidates, following the procedure of [14]. For measurement, we use the average of the pairwise distance between candidates as below.

$$\text{Diversity}(\mathcal{D}) = \frac{1}{|\mathcal{D}|(|\mathcal{D}| - 1)} \sum_{\mathbf{x} \in \mathcal{D}} \sum_{\mathbf{x}' \in \mathcal{D} \setminus \{\mathbf{x}\}} d(\mathbf{x}, \mathbf{x}') \tag{11}$$

where $d(\mathbf{x}, \mathbf{x}')$ is a pairwise distance between samples. For discrete tasks, we use the hamming-ball distance metric. For continuous tasks, we compute L2 distance.

Table 13 illustrates the effect of filtering on performance and diversity. As expected, we achieve higher performance through filtering while sacrificing the diversity of the candidate set. It might be beneficial to automatically balance performance and diversity trade-off by measuring the uncertainty of the proxy function. We leave it as a future work.

Table 13: Impact of filtering on performance and diversity of designs

| Method | TFBind8 | | Ant | | D'Kitty | |
|---|---|---|---|---|---|---|
| | Performance | Diversity | Performance | Diversity | Performance | Diversity |
| GTG | **0.976 ± 0.020** | 1.13 ± 0.03 | **0.963 ± 0.009** | 9.41 ± 1.96 | **0.971 ± 0.009** | 0.41 ± 0.07 |
| GTG w/o Filtering | 0.920 ± 0.036 | **1.17 ± 0.01** | 0.952 ± 0.026 | **17.02 ± 3.56** | 0.965 ± 0.007 | **0.73 ± 0.06** |

### D.2.3 Impact of Exploration Level ($\alpha$)

In this section, we explore the impact of the exploration level ($\alpha$) on the generated samples. As depicted in Figure 3c, increasing $\alpha$ leads to higher performance, indicating the importance of classifier-free guidance. However, we observe that conditioning on extremely high $\alpha$ leads to sub-optimal performance, as illustrated in Figure 9. Conditioning on extremely high $\alpha$ guides the diffusion model to over-exploration, resulting in sub-optimal out-of-distribution designs. Note that we do not fine-tune $\alpha$ for each task and fix it with the value of 0.8 across all tasks, which generally exhibits good performance.

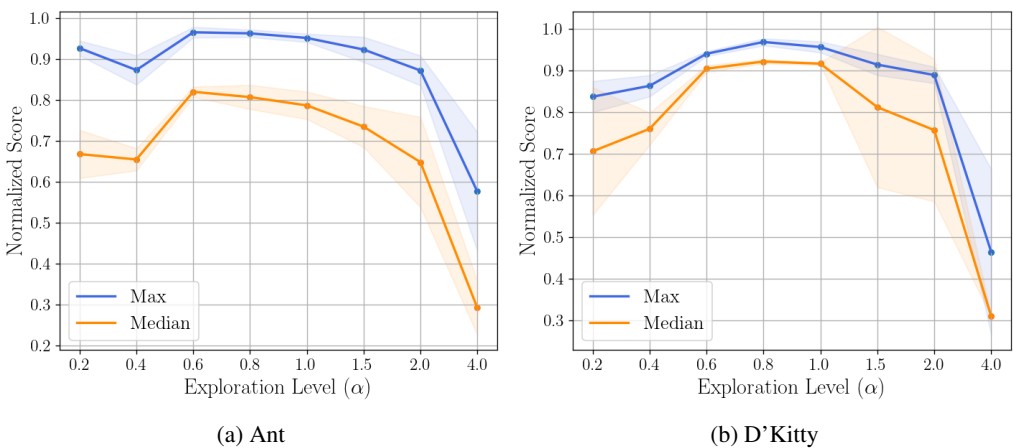

(a) Ant                           (b) D'Kitty

Figure 9: Performance of GTG in Ant and D'Kitty tasks with extremely high $\alpha$ values

### D.2.4 Assumption on $y^*$

We assume that the optimal value $y^*$ of each task is known, following prior works [13, 16]. However, it is not always possible to know the exact optima. To this end, we estimate $y^*$ with $\gamma \cdot y_{max}$, where $y_{max}$ is the maximum value of the dataset and evaluate GTG by conditioning on the estimated value. As depicted in Table 14, conditioning on $\gamma \cdot y_{max}$ achieves comparable performance and even outperforms the performance of conditioned on exact optima in the TFBind8 task. However, it introduces an additional hyperparameter $\gamma$, whose optimal value varies across tasks. Therefore, we rely on assuming the exact optima, which is not an issue in many problems.

Table 14: Analysis on relaxing assumption of known $y^*$.

| **Method** | TFBind8 | TFBind10 | Superconductor | Ant | D'Kitty |
|---|---|---|---|---|---|
| $\gamma = 1.0$ | $0.973 \pm 0.020$ | $0.687 \pm 0.122$ | $0.490 \pm 0.055$ | $0.898 \pm 0.027$ | $0.965 \pm 0.011$ |
| $\gamma = 1.5$ | $\mathbf{0.984 \pm 0.010}$ | $0.684 \pm 0.123$ | $0.494 \pm 0.052$ | $0.960 \pm 0.010$ | $0.947 \pm 0.012$ |
| $\gamma = 2.0$ | $0.976 \pm 0.020$ | $0.684 \pm 0.123$ | $0.490 \pm 0.046$ | $0.957 \pm 0.011$ | $0.925 \pm 0.022$ |
| $y^*$ is known | $0.976 \pm 0.020$ | $\mathbf{0.698 \pm 0.127}$ | $\mathbf{0.519 \pm 0.045}$ | $\mathbf{0.963 \pm 0.009}$ | $\mathbf{0.971 \pm 0.009}$ |

### D.3 Effect of Unsupervised Pretraining

It might be beneficial to pretrain the diffusion model with unlabeled data when we have limited data points. Specifically, there is a recent work EXPT [29], which trains an autoregressive model using synthetic trajectories constructed from the large-scale unlabeled dataset and adapts new tasks by conditioning on a few labeled points. To this end, we discuss the effect of pre-training GTG with unlabeled datasets. We follow a similar procedure of EXPT to generate a synthetic dataset. Formally, we sample synthetic functions from Gaussian Processes [46] with an RBF kernel and assign pseudo values to the unlabeled data points from synthetic functions. Please refer to [29] for a more detailed setting. Given a synthetic dataset, we pretrain diffusion models with trajectories constructed from the dataset using the proposed method. Then, we generate samples by conditioning on context data points from the labeled dataset. For labeled dataset, we randomly select 1% of the original dataset.

Table 15 shows the experiment results on various Design-Bench tasks. As shown in the table, pretraining generally improves the performance of GTG in the sparse data setting. We also find that GTG with pretraining outperforms ExPT in 3 of 5 tasks. While we do not assume the existence of the large-scale unlabeled dataset in the main experiment and pretraining is not a main focus of our research, it might be beneficial to analyze the effect of pretraining with synthetic datasets in offline MBO thoroughly as in other problems [47, 48].

Table 15: Impact of pretraining with a synthetic dataset on performance. Experiments are conducted with three random seeds.

| Method | TFBind8 | TFBind10 | Superconductor | Ant | D'Kitty |
|---|---|---|---|---|---|
| ExPT | 0.837 ± 0.036 | 0.635 ± 0.036 | 0.471 ± 0.030 | **0.955 ± 0.021** | **0.961 ± 0.006** |
| GTG | 0.948 ± 0.009 | 0.666 ± 0.051 | 0.526 ± 0.032 | 0.655 ± 0.051 | 0.949 ± 0.013 |
| GTG w Pretraining | **0.953 ± 0.030** | **0.703 ± 0.018** | **0.564 ± 0.038** | 0.897 ± 0.015 | 0.930 ± 0.005 |

### D.4 Time Complexity of Sampling Procedure

In this section, we analyze the time complexity of the sampling procedure of GTG. To generate trajectories, we run $T = 200$ denoising timesteps with classifier-free guidance and context-conditioning to sample $N = 128$ trajectories, which takes approximately 9.41s and 9.47s in wall clock time for the Ant and D'Kitty tasks, respectively. We visualize the trade-off between the performance and runtime of sampling by varying the number of denoising timesteps. As shown in Figure 10, we can decrease the number of denoising timesteps even one-tenth with minimal loss in performance. Please note that sampling time is negligible compared to evaluating black-box functions, which is mostly expensive in real-world settings.

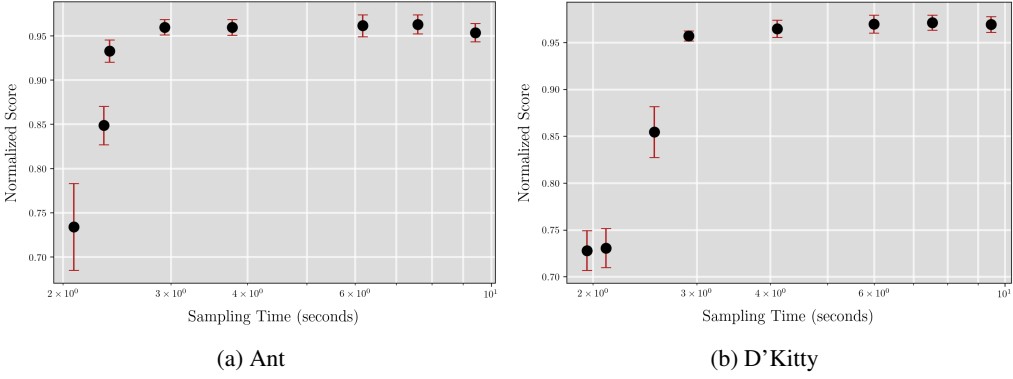

      (a) Ant               (b) D'Kitty

Figure 10: Trade-off between Performance and Sampling time in Ant and D'Kitty tasks.

## D.5 Extended Experiment Results

In this section, we present extended experiment results in sparse and noisy datasets. As shown in Tables 16 and 17, our method outperforms most baselines in various practical settings. Note that we cannot conduct experiments with NEMO and RoMA, as there is no code publicly available.

Table 16: Experiments on Sparse Datasets.

| Method | TFBind8 | | | Dkitty | | |
|---|---|---|---|---|---|---|
| | 1% | 20% | 50% | 1% | 20% | 50% |
| BO-qEI | 0.878 ± 0.048 | 0.878 ± 0.082 | 0.863 ± 0.085 | 0.884 ± 0.001 | 0.891 ± 0.005 | 0.891 ± 0.003 |
| CMA-ES | 0.879 ± 0.066 | 0.920 ± 0.039 | 0.927 ± 0.037 | 0.722 ± 0.002 | 0.723 ± 0.001 | 0.723 ± 0.001 |
| REINFORCE | 0.945 ± 0.036 | 0.936 ± 0.027 | 0.910 ± 0.032 | 0.615 ± 0.178 | 0.614 ± 0.176 | 0.360 ± 0.130 |
| Grad Ascent | 0.897 ± 0.060 | 0.954 ± 0.037 | 0.951 ± 0.027 | 0.610 ± 0.172 | 0.822 ± 0.043 | 0.868 ± 0.016 |
| COMs | 0.941 ± 0.032 | 0.954 ± 0.029 | 0.969 ± 0.016 | 0.918 ± 0.005 | 0.915 ± 0.057 | 0.958 ± 0.015 |
| BDI | 0.898 ± 0.000 | 0.952 ± 0.000 | **0.988 ± 0.000** | 0.865 ± 0.000 | 0.927 ± 0.000 | 0.938 ± 0.000 |
| ICT | 0.899 ± 0.045 | 0.925 ± 0.035 | 0.962 ± 0.019 | 0.946 ± 0.010 | 0.949 ± 0.010 | 0.954 ± 0.008 |
| CbAS | 0.908 ± 0.043 | 0.915 ± 0.036 | 0.909 ± 0.040 | 0.887 ± 0.016 | 0.895 ± 0.010 | 0.900 ± 0.008 |
| MINs | 0.871 ± 0.083 | 0.882 ± 0.021 | 0.935 ± 0.027 | 0.926 ± 0.008 | 0.941 ± 0.008 | 0.938 ± 0.007 |
| DDOM | 0.851 ± 0.082 | 0.906 ± 0.050 | 0.896 ± 0.048 | 0.938 ± 0.007 | 0.945 ± 0.011 | 0.944 ± 0.008 |
| BONET | 0.791 ± 0.079 | 0.824 ± 0.061 | 0.884 ± 0.072 | 0.875 ± 0.004 | 0.939 ± 0.007 | 0.940 ± 0.009 |
| PGS | 0.914 ± 0.043 | 0.866 ± 0.064 | 0.896 ± 0.100 | 0.939 ± 0.023 | 0.952 ± 0.022 | 0.963 ± 0.023 |
| **GTG (Ours)** | **0.948 ± 0.009** | **0.964 ± 0.025** | 0.973 ± 0.016 | **0.949 ± 0.013** | **0.957 ± 0.009** | **0.968 ± 0.002** |

Table 17: Experiments on Noisy Datasets.

| Method | TFBind8 | | | Dkitty | | |
|---|---|---|---|---|---|---|
| | 1% | 20% | 50% | 1% | 20% | 50% |
| BO-qEI | 0.744 ± 0.089 | 0.716 ± 0.091 | 0.579 ± 0.114 | 0.891 ± 0.003 | 0.891 ± 0.012 | 0.884 ± 0.000 |
| CMA-ES | 0.968 ± 0.011 | 0.961 ± 0.013 | 0.876 ± 0.061 | 0.863 ± 0.022 | 0.852 ± 0.014 | 0.839 ± 0.012 |
| REINFORCE | 0.825 ± 0.054 | 0.879 ± 0.041 | 0.819 ± 0.051 | 0.409 ± 0.171 | 0.560 ± 0.194 | 0.619 ± 0.193 |
| Grad Ascent | 0.955 ± 0.022 | 0.938 ± 0.025 | 0.917 ± 0.034 | 0.911 ± 0.009 | 0.856 ± 0.024 | 0.830 ± 0.047 |
| COMs | 0.928 ± 0.028 | 0.938 ± 0.040 | 0.915 ± 0.057 | 0.936 ± 0.009 | 0.926 ± 0.012 | 0.925 ± 0.013 |
| BDI | **0.980 ± 0.005** | 0.886 ± 0.051 | 0.873 ± 0.048 | 0.929 ± 0.008 | 0.908 ± 0.010 | 0.918 ± 0.016 |
| ICT | 0.941 ± 0.013 | 0.950 ± 0.023 | 0.921 ± 0.054 | 0.940 ± 0.029 | 0.914 ± 0.024 | 0.896 ± 0.000 |
| CbAS | 0.916 ± 0.041 | 0.916 ± 0.034 | 0.906 ± 0.033 | 0.901 ± 0.009 | 0.898 ± 0.017 | 0.888 ± 0.013 |
| MINs | 0.885 ± 0.057 | 0.947 ± 0.032 | 0.883 ± 0.068 | 0.941 ± 0.006 | 0.938 ± 0.008 | 0.932 ± 0.008 |
| DDOM | 0.896 ± 0.048 | 0.887 ± 0.065 | 0.887 ± 0.065 | 0.944 ± 0.009 | 0.945 ± 0.011 | 0.926 ± 0.020 |
| BONET | 0.904 ± 0.044 | 0.822 ± 0.113 | 0.773 ± 0.143 | 0.942 ± 0.008 | 0.927 ± 0.024 | 0.924 ± 0.010 |
| PGS | 0.906 ± 0.030 | 0.911 ± 0.033 | 0.869 ± 0.039 | 0.942 ± 0.005 | 0.923 ± 0.009 | 0.891 ± 0.016 |
| **GTG (Ours)** | 0.976 ± 0.015 | **0.967 ± 0.026** | **0.948 ± 0.029** | **0.955 ± 0.008** | **0.947 ± 0.015** | **0.937 ± 0.013** |

## D.6 Additional Visualization on Toy 2D Experiment

We present additional visualization results from the Toy 2D experiment. As shown in Figure 11, GTG is able to generate diverse trajectories toward high-scoring designs by conditioning on different context points and classifier-free guidance.

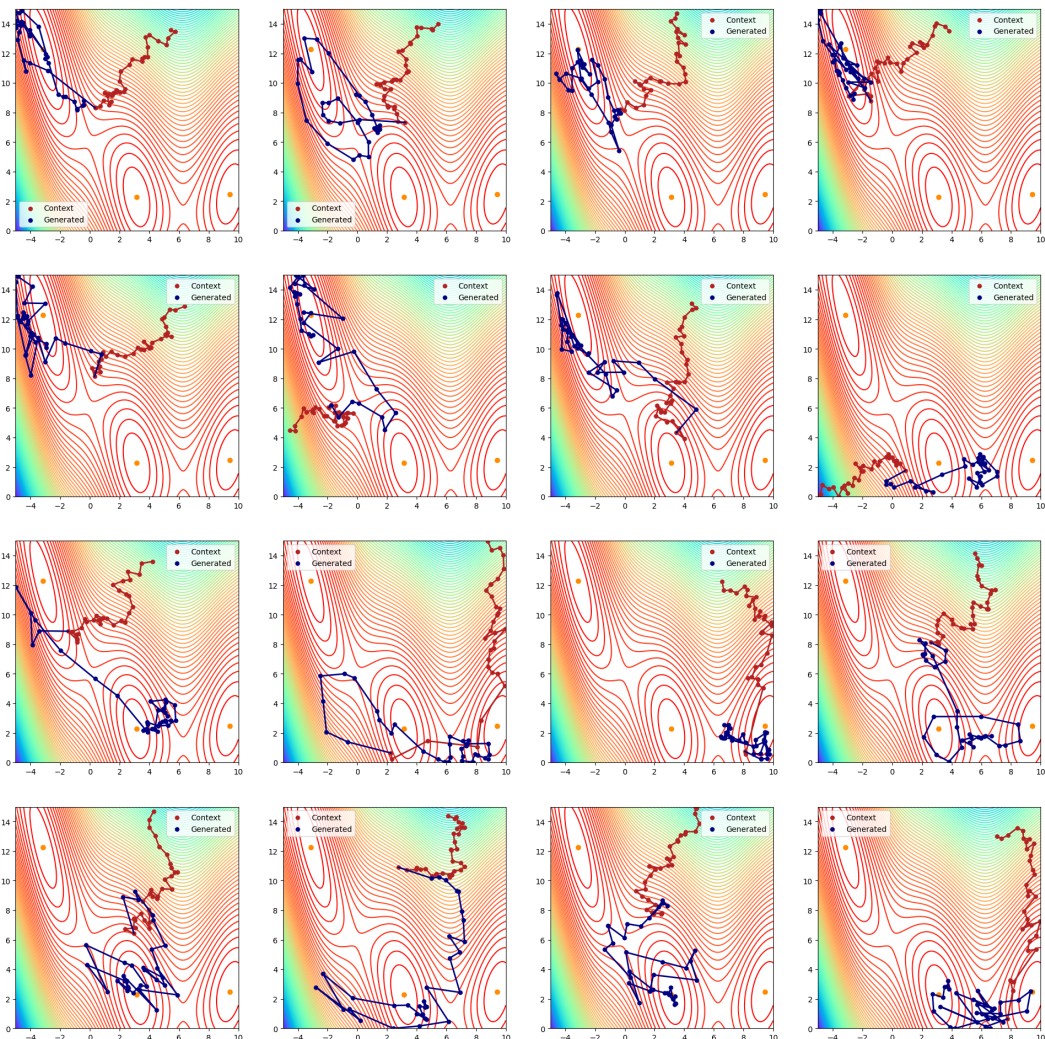

Figure 11: Extra visualization of generated trajectories with GTG in Branin Task.

# E Broader Impact

Optimization for real-world designs presents both opportunities and risks. For instance, while the design of new pharmaceuticals holds the promise of curing previously untreatable diseases, there is the potential for misuse, such as creating harmful biochemical agents. Researchers should be diligent to ensure that their innovations are employed in ways that contribute positively to societal welfare.

