# OpenReview forum: "Guided Trajectory Generation with Diffusion Models for Offline Model-based Optimization"
_NeurIPS.cc/2024/Conference — NeurIPS 2024 poster_

### Official Review · Reviewer_FGy7 · 2024-07-10

**Soundness:** 2
**Presentation:** 3
**Contribution:** 2
**Rating:** 5
**Confidence:** 3

**Summary:**

The paper proposes a novel conditional generative modeling approach using diffusion models for offline model-based optimization (MBO). The method constructs synthetic trajectories toward high-scoring regions, trains a conditional diffusion model, and samples multiple trajectories to explore and select high-fidelity designs.

**Strengths:**

1.	Introduces a new approach to generating trajectories that consistently move towards high-scoring regions.
2.	Leverages the powerful capabilities of diffusion models to handle complex and high-dimensional data.
3.	The method utilizes classifier-free guidance and context conditioning during the sampling process, enhancing the exploration of high-scoring regions.

**Weaknesses:**

1.	The authors have adopted the Diffusion model for offline model-based optimization, which is a current trending approach. It is recommended that the authors emphasize their unique contributions and clarify the distinctions between their method and previous approaches.
2.	The model filters candidates through a trained proxy model. The authors should elaborate on how data handling is managed during experiments to ensure that there is no label leakage and discuss whether the comparisons with other models are conducted fairly.
3.	The authors should provide a more detailed description and discussion of the experimental results.

**Questions:**

Please see weaknesses

---

> ### Author Rebuttal · Authors · 2024-08-06
>
> Thank you for the positive feedback and further suggestions that could enhance our manuscript.
>
> > (Weakness 1) The authors have adopted the Diffusion model for offline model-based optimization, which is a current trending approach. It is recommended that the authors emphasize their unique contributions and clarify the distinctions between their method and previous approaches.
>
> We acknowledge that several concurrent works adopt diffusion models for offline model-based optimization. DDOM [1] is the pioneering work utilizing diffusion models for offline MBO. Following DDOM, DEMO [2] trains a diffusion model to match a pseudo-target distribution constructed by gradient ascent and uses the model to edit designs in the offline dataset. Diff-Opt [3] considers a constrained optimization setting and introduces a two-stage framework that begins with a guided diffusion process for warm-up, followed by a Langevin dynamics stage for further correction. Diff-BBO [4] measures the uncertainty of generated designs to select the optimal target value for conditioning the diffusion model.
>
> Our method uniquely distinguishes itself from prior and concurrent works by using diffusion models to generate trajectories toward high-scoring regions, learning to improve solutions from the dataset. While we mention this part in the related work section, we will modify the manuscript to emphasize our unique contributions in the introduction.
>
> > (Weakness 2) The model filters candidates through a trained proxy model. The authors should elaborate on how data handling is managed during experiments to ensure that there is no label leakage and discuss whether the comparisons with other models are conducted fairly.
>
> In the filtering stage, we train our proxy model with only offline datasets for fair comparison.
>
> > (Weakness 3) The authors should provide a more detailed description and discussion of the experimental results.
>
> Sorry for missing detailed description and discussion on the experiment results. We will provide more thorough analysis on our main experiment in the manuscript as written below.
>
> Our experiment results generally surpass forward approaches, which struggle to fall into OOD designs, especially in high-dimensional settings. We also observe that our method outperforms inverse approaches, including DDOM, which utilizes the diffusion model. It demonstrates that generating trajectories towards high-scoring regions can be more effective than generating a single design, as we can distill the knowledge of the landscape of the target function into the generator. Our method achieves higher performance compared to BONET, which also generates trajectories. It indicates that our novel trajectory construction strategy effectively guides the diffusion model to explore diverse paths toward high-scoring regions.
>
> [1] Krishnamoorthy, Siddarth, Satvik Mehul Mashkaria, and Aditya Grover. "Diffusion models for black-box optimization." International Conference on Machine Learning. PMLR, 2023.
>
> [2] Yuan, Ye, et al. "Design Editing for Offline Model-based Optimization." arXiv preprint arXiv:2405.13964 (2024).
>
> [3] Kong, Lingkai, et al. "Diffusion models as constrained samplers for optimization with unknown constraints." arXiv preprint arXiv:2402.18012 (2024).
>
> [4] Wu, Dongxia, et al. "Diff-BBO: Diffusion-Based Inverse Modeling for Black-Box Optimization." arXiv preprint arXiv:2407.00610 (2024).

---

### Official Review · Reviewer_nEhC · 2024-07-11

**Soundness:** 3
**Presentation:** 3
**Contribution:** 2
**Rating:** 6
**Confidence:** 4

**Summary:**

The paper introduces Guided Trajectory Generation (GTG) for offline model-based optimization (MBO). GTG creates synthetic trajectories from offline data, using locality bias to ensure consistent improvement. A conditional diffusion model generates these trajectories based on their scores. The method then uses guided sampling and a proxy function to identify the best designs. GTG shows superior performance on the Design-Bench benchmark compared to other methods.

**Strengths:**

- The authors effectively motivate their approach with a toy 2D Branin example, highlighting the differences from previous synthetic trajectory-based methods.
- Extensive ablation studies demonstrate the robustness and effectiveness of the proposed method.
- The paper is well-written and easy to follow, with a straightforward and effectively communicated idea.

**Weaknesses:**

- TFBind8 and TFBind10 are very similar. It would be better to see experiments on more varied discrete benchmarks like NAS. Also, why do the results for TFBind10 have such high variance?
- It’s not clear how this approach deals with out-of-distribution (OOD) problems. Diffusion models usually generate in-distribution data, so where does the robustness against OOD desings come from?
- The proxy function uses the standard regression approach, which can be quite fragile when estimating OOD designs. Have you investigated using COMS or ROMA models as a proxy?
- There isn't a principled mathematical explanation for why this approach works.
- The paper doesn’t include 50th percentile results, which are a standard part of evaluation in this field.

**Questions:**

- Please check the weaknesses
- For the discrete tasks, Is a latent space used or is everything kept discrete during diffusion and proxy training?
- How many gradient steps are used for gradient ascent methods like COMs and RoMA? Is it 64? And is the trajectory length the same for BONET and PGS?
- Were the hyperparameters picked using the proxy or the oracle?
- Are designs picked at the end of the generation process, or can they be selected from the middle of the trajectory? Is the filtering applied to results generated by BONET or PGS? How does this affect their outcomes?
- How many designs are initially considered, and how many are selected at the end (128 out of how many)?
- Why is the perturbation epsilon different for DKitty?

**Limitations:**

The authors address the limitations and potential negative impacts in their paper.

---

> ### Author Rebuttal · Authors · 2024-08-06
>
> Thank you for the critical reviews, insightful feedback, and further suggestions to enhance our manuscript.
>
> > (Weakness 1) More varied discrete benchmarks and high variance in TFBind10 results
>
> We excluded NAS as it takes too long to evaluate. Instead, to verify the effectiveness of our method on various discrete tasks, we conduct additional experiments on UTR. We use the reported scores in PGS for other baselines. As shown in Table 19 attached in PDF, we achieve the second-best results in the UTR task.
>
> For high variance in TFBind10, we observe that it is due to the variability of the seeds. When we conduct experiments on more seeds (16 seeds), we find that our method still showed high performance with lower variance.
>
> **Performance of GTG on TFBind10 with more random seeds.**
> || TFBind10|
> | -|:-|
> | GTG (8 seeds)  | 0.698 ± 0.127 |
> | GTG (16 seeds) | 0.699 ± 0.091 |
>
> > (Weakness 2) It’s not clear how this approach deals with out-of-distribution (OOD) problems.
>
> Thank you for your interest in how our method addresses out-of-distribution (OOD) problems. We introduce classifier-free guidance to guide the diffusion model towards exploring high-scoring regions. To mitigate the OOD issue, we introduce $\alpha$, which controls the exploration level of the generated trajectories. We find that setting $\alpha=0.8$ generally exhibits good performance while high $\alpha$ results in sub-optimal OOD designs. Please refer to Section 5 and Appendix D.2 for a more discussion on the impact of $\alpha$.
>
> > (Weakness 3) Using COMS or ROMA models as a proxy
>
> As you mentioned, the proxy function can be fragile when estimating OOD designs. To mitigate this issue, we use ensembles of MLP as a proxy with rank-based reweighting during training to focus on high-scoring regions as written in Appendix B.2. Nevertheless, using COMs and ROMA is a promising approach to enhance the robustness against OOD designs.
>
> To this end, we replace proxy with COMs and ROMA and evaluate the performance of our method on TFBind8 and Dkitty tasks. As shown in Table 19 attached in PDF, we find no significant difference in performance based on the choice of different proxy functions.
>
> > (Weakness 4) Absence of principled mathematical explanation
>
> While there is no principled mathematical explanation, our method shows promising results across various real-world tasks. We also conduct a thorough analysis of our method using Toy 2D experiment visualizations in Figure 2. Additionally, we perform experiments on practical settings such as sparse and noisy datasets, demonstrating the versatility of our method.
>
> > (Weakness 5) Absence of median performance
>
> We apologize for missing the median performance results. We will update the manuscript with median performance results. As shown in the table 20 attached in the PDF, we also achieve the highest mean rank (4.0) in terms of the median performance.
>
> > (Question 1) For the discrete tasks, Is a latent space used, or is everything kept discrete during diffusion and proxy training?
>
> As mentioned in Appendix A.2, we convert discrete input into a continuous vector and adopt a continuous diffusion and proxy model.
>
> > (Question 2) How many gradient steps are used for gradient ascent methods like COMs and RoMA? Is it 64? And is the trajectory length the same for BONET and PGS?
>
> We strictly follow the original procedures of the baselines written in their papers and publicly available code. For BONET, while the original paper generates 4 trajectories with a prediction length of 64, we generate 4 trajectories with a prediction length of 32 to match the evaluation budget of 128.
>
> > (Question 3) Are designs picked at the end of the generation process, or can they be selected from the middle of the trajectory?
>
> We aggregate all designs generated by trajectories and select candidates for evaluation with a filtering procedure.
>
> > (Question 4) Is the filtering applied to results generated by BONET or PGS? How does this affect their outcomes?
>
> Both methods do not apply a filtering strategy after generation. While PGS generates synthetic trajectories, it ultimately selects final candidates using a policy trained with the trajectories. Therefore, we examine the effect of filtering only in BONET. As shown in the table, filtering can improve the performance of BONET. However, we find that our method still outperforms BONET in terms of performance.
>
> **Performance of BONET with filtering strategy.**
> | | TFBind8| Dkitty|
> |:- |:- |:-|
> | BONET| 0.831 ± 0.109 | 0.950 ± 0.014 |
> | BONET + Filtering | 0.913 ± 0.122 | 0.954 ± 0.011 |
> | GTG (Ours) | 0.976 ± 0.020 | 0.971 ± 0.009 |
>
> > (Question 5) How many designs are initially considered, and how many are selected at the end (128 out of how many)?
>
> As mentioned in Section 4.4, we sample $N=128$ trajectories conditioning on $C=32$ context data points, so overall $4,096$ designs are initially considered. Then $Q=128$ designs are finally selected. As sampling procedure can be conducted in parallel, time complexity is not too problematic. For time complexity of sampling procedure, please refer to Appendix D.4.
>
> > (Question 6) Why is the perturbation epsilon different for DKitty?
>
> We introduce perturbation epsilon ($\epsilon$) to prevent synthetic trajectories from converging into a single maxima in the offline dataset. We use small $\epsilon$ for Dkitty as there are relatively many samples with high scores in the offline dataset compared to other datasets.
>
> We also observe that even with the same $\epsilon=0.05$ for the Dkitty task, there is no big difference in performance, and it still outperforms other baselines, as shown in the table. For analysis on $K$ and $\epsilon$ for constructing trajectories, please refer to Appendix D.1.
>
> **Performance of GTG on DKitty task with different perturbation epsilon ($\epsilon$).**
> || D'Kitty|
> | -- |:--- |
> | ICT (Best among baselines) | 0.960 ± 0.014 |
> | GTG ($\epsilon=0.01$)| 0.971 ± 0.009 |
> | GTG ($\epsilon=0.05$)| 0.965 ± 0.008 |

---

> > ### Comment · Reviewer_nEhC · 2024-08-13
> >
> > Thank you for your answers. I have raised my rating.

---

> > > ### Author Response · Authors · 2024-08-13
> > >
> > > Thank you for your kind response. If there are any remaining issues of discussion, please feel free to share them with us. We are always ready to engage in further discussion. Once again, we appreciate your thoughtful feedback.

---

### Official Review · Reviewer_MZpe · 2024-07-13

**Soundness:** 3
**Presentation:** 3
**Contribution:** 2
**Rating:** 5
**Confidence:** 4

**Summary:**

The paper consider the problem of offline optimization where the goal is to find the optima of a black-box function in a zero-shot manner without online evaluations. The key idea is to generate trajectories with a locality biased heuristic and employ a conditional diffusion model to learn the distribution of these trajectories. Experiments are performed on multiple tasks from design bench benchmark.

**Strengths:**

- I think looking at construction of trajectories is a really interesting problem in the context of offline optimization. Other than the two references mentioned in the paper, another recent ICML paper (listed below) also uses monotonic trajectories in the context of this problem that shows this is a recurring theme.

    Hoang, M., Fadhel, A., Deshwal, A., Doppa, J., & Hoang, T. N. (2024). Learning Surrogates for Offline Black-Box Optimization via Gradient Matching. In Forty-first International Conference on Machine Learning.

- I like the ablation experiments in the experimental section that shows the efficacy of different components of the proposed approach.

**Weaknesses:**

- This is not specific to the paper as such but the tasks in design-bench benchmark have multiple issues and as a result, some of the results are not reliable. For example, the offline dataset in superconductor task has multiple copies of the same inputs but with different outputs. As a result, the random forest oracle which is fit on this offline data is not reliable.

- The paper calls existing trajectory generation approaches (like sorted monotonic trajectories) as heuristics but the proposed approach is also a heuristic. Is there more principled/theoretical explanation for the proposed approach?

**Questions:**

Please see weaknesses section.

**Limitations:**

Please see weaknesses section.

---

> ### Author Rebuttal · Authors · 2024-08-06
>
> Thank you for the valuable review and positive feedback!! As only two questions have been raised by the reviewer, we will address it in this response. If you have any additional questions, please do not hesitate to let us know!
>
> > (Weakness 1) Unreliable results of superconductor task.
>
> We acknowledge the issues with Design-Bench, particularly in the Superconductor task. However, we have found that by fixing the seed for evaluation, we can obtain consistent outputs for multiple copies of the same inputs. Please note that we have reproduced baselines whose code is publicly available using the same evaluation procedure for a fair comparison.
>
> > (Weakness 2) The paper calls existing trajectory generation approaches (like sorted monotonic trajectories) as heuristics but the proposed approach is also a heuristic. Is there more principled/theoretical explanation for the proposed approach?
>
> We apologize for overstating our proposed method. Our trajectory construction method can also be considered a heuristic since it does not include learning components. Nevertheless, we focus on constructing trajectories that give us more valuable information for learning to improve designs toward diverse high-scoring regions. We have empirically found that constructing trajectories with locality bias can guide a diffusion model to explore high-scoring regions in the Design-Bench and its variants. Furthermore, we also find that the computational time for constructing trajectories does not require a significantly large amount of time, as shown in Table 9 in Appendix B.1.

---

> > ### Comment · Reviewer_MZpe · 2024-08-11
> > **Response to Rebuttal**
> >
> > Thanks for your response to my questions. I am fairly convinced that this will be a good addition to the offline model based optimization literature. I would request to add a discussion about following things:
> >
> > - I appreciate the special focus on fixing the seed and reproducing the baselines but the Superconductor task has one specific issue that it has multiple copies of the same inputs but with different outputs. As a result, the oracle is not reliable even though we can reproduce the results. Please add this while presenting the results about this task.
> >
> > - Hoang, M., Fadhel, A., Deshwal, A., Doppa, J., & Hoang, T. N. (2024). Learning Surrogates for Offline Black-Box Optimization via Gradient Matching. In Forty-first International Conference on Machine Learning. is another recent paper that constructs trajectories. It will be good to add discussion about this paper in the related work section.

---

> > > ### Author Response · Authors · 2024-08-11
> > >
> > > Thank you for your positive feedback! We will incorporate an explanation of the unreliability of the Superconductor task and a discussion about the recent paper by Hoang et al. (2024) into our manuscript.
> > >
> > > Please let me know if you have any further suggestions or adjustments you would like us to consider.

---

### Official Review · Reviewer_gJu1 · 2024-07-13

**Soundness:** 3
**Presentation:** 3
**Contribution:** 3
**Rating:** 7
**Confidence:** 3

**Summary:**

This paper introduces Guided Trajectory Generation (GTG), a novel conditional generative modeling approach to solve the MBO problem. GTG consists of three parts, including trajectories construction, model training and trajectories sampling and filtering. Experimental results on various tasks, including a toy 2D experiment and Design-Bench variants, demonstrate the method's effectiveness.

**Strengths:**

1. This paper is well-written with a great motivation and description. Especially, authors introduce the conditional guidance to explore high-scoring regions.
2. The effectiveness of the proposed method is verified through experiments on multiple tasks and a detailed comparison with existing methods is presented.
3. The paper analyzes the hyperparameter settings and explores the effects of different hyperparameter settings on the model performance.

**Weaknesses:**

1. The method proposed is similar to Decision Diffuser [1]. The method proposed focuses on the offline model-based optimization problem while Decision Diffuser is used to sovle the offline RL problem.
2. It is not clear that in loss function, only x is used to calculate loss or both x and y are used.
3. In this paper, the concept of filtering in Section 3.4 and Section 5 are not clarified clearly.

[1] Anurag Ajay, Yilun Du, Abhi Gupta, Joshua Tenenbaum, Tommi Jaakkola, and Pulkit Agrawal. Is conditional generative modeling all you need for decision-making? arXiv preprint arXiv:2211.15657, 2022.

**Questions:**

1. Please clarify the difference between the method proposed in this paper and Decision Diffuser [1].
2. In Section 3.4 and Appendix D.3, the distribution of data used to train models must be close to the real distribution. Otherwise, out of distribution could occur, leading to an error evaluation of trajectories.
3. Please clarify the concept of filtering in Section 3.4 and Section 5. Whether they are the same operation or not?

[1] Anurag Ajay, Yilun Du, Abhi Gupta, Joshua Tenenbaum, Tommi Jaakkola, and Pulkit Agrawal. Is conditional generative modeling all you need for decision-making? arXiv preprint arXiv:2211.15657, 2022.

**Limitations:**

Yes, the authors have discussed the limitations and potential negative societal impact of their work.

---

> ### Author Rebuttal · Authors · 2024-08-06
>
> Thank you for the insightful review and positive feedback!
> > (Weakness 1 & Question 1) Clarification on the difference between the proposed method and Decision Diffuser.
>
> We would like to highlight that our method aims to find a design that maximizes the target black-box function while Decision Diffuser is a planner to solve offline RL problems by generating high-rewarding trajectories that follow environment dynamics.
>
> Instead of utilizing diffusion model to generate a single design, we train diffusion model with synthetic trajectories constructed from the dataset. While several prior works construct trajectories from datasets, we developed a novel method for constructing trajectories with locality bias to help generator better understand the landscape of the target function. As reviewer MZpe mentioned, how to construct trajectories is a interesting problem in the context of MBO. The ablation study summarized in Table 4 underscores that our method outperforms prior strategies across various tasks.
> > (Weakness 2) It is not clear that in loss function, only x is used to calculate loss or both x and y are used.
>
> We apologize for the confusion in the notation. The trajectory $\tau$, which is a set of $(x, y)$ input pairs, should be used to calculate the loss in line 5 of Algorithm 2. We train diffusion model to generate trajectories, not a single design $x$. I will modify the manuscript to clarify this.
>
> > (Question 2) In Section 3.4 and Appendix D.3, the distribution of data used to train models must be close to the real distribution. Otherwise, out of distribution could occur, leading to an error evaluation of trajectories.
>
> For both Section 3.4 and Appendix D.3, we train proxy model with only offline dataset. As our objective is filtering high-fildelity designs with the proxy, we introduce a rank-based reweighting suggested by [1] during training to make the proxy model focus on high-scoring regions. To prevent high error in evaluation, we train ensembles of MLP for robustness of the proxy model. For more details, please refer to Appendix B.2 and our code.
>
> [1] Austin Tripp, Erik Daxberger, and José Miguel Hernández-Lobato. Sample-efficient optimization in the latent space of deep generative models via weighted retraining. Advances in Neural Information Processing Systems, 33:11259–11272, 2020.
>
> > (Weakness 3 & Question 3) Clarification of the concept of filtering in Section 3.4 and Section 5
>
> We apologize for the confusion regarding the concept of filtering in the manuscript. Both Section 3.4 and Section 5 refer to the same concept. Among samples from multiple generated trajectories, we select candidates for evaluation by filtering with the proxy function.

---

### Author Rebuttal · Authors · 2024-08-06

We sincerely thank the review committee for their detailed feedback. We appreciate the recognition of our paper's strengths, highlighted by the reviewers: **well-written** (gJu1, nEhc), **novel** (nEhc, FGy7), and **extensive experiments and ablations** (gJu1, MZpe, nEhc). In response to the reviewers' feedback, we provide a summary of conducted additional experiments:

- **More discrete tasks:** We conduct additional experiments on UTR tasks to demonstrate the effectiveness of our method on various discrete tasks. We achieved the second-best results on the UTR task.

- **Different proxy functions**: We conduct additional experiments with different proxy functions such as COMs and ROMa for filtering strategy. We find that there is no significant difference in performance depending on the choice of proxy function.

- **Median Performance**: We report the median score of our method and all baselines. We achieve the highest mean rank (4.0) regarding median performance.

Detailed responses have been provided for each point raised by the reviewers.

---

### Author Response · Authors · 2024-08-12

Dear Reviewers,

We wanted to remind you that the author-reviewer discussion period ends in two days. We value your feedback and kindly request your response to our rebuttal.

Your participation will significantly help us enhance our work. If you have any questions or need more information, please don't hesitate to ask.

We appreciate your time and look forward to your input.

Best regards,

The Authors

---

### Decision · Program_Chairs · 2024-09-25

**Decision:**

Accept (poster)

**Comment:**

This paper is concerned with model-based optimization (MBO). The authors propose to leverage the diffusion models with guided sampling methods to synthesize trajectories to improve the solutions. Overall, I think the authors make a good contribution to MBO as they successfully demonstrate that the diffusion models can be used as a trajectory synthesizer for MBO tasks, which will motivate more studies on along this direction. The authors also provide good responses to ease the concerns from the reviewers. As a result, I recommend acceptance of this paper.